# OpenReview forum: "GhostWord: A Fine-Grained Backdoor Attack on Automatic Speech Recognition"
_TMLR — Under review for TMLR_

### Review · Reviewer_cqPp · 2026-06-10

**Summary Of Contributions:**

This paper studies data-poisoning backdoor attacks against ASR and argues that the dominant "phrase-level" recipe -- one fixed acoustic trigger mapped to one fixed target transcript -- is fragile because it leaves obvious artifacts (many identical transcripts, triggers parked in non-speech regions) that trivial preprocessing can strip out. The authors propose GhostWord, a word-level, time-localized alternative. The attacker builds a set of codebooks, each pairing a short ($\approx 400$ ms) additive-noise trigger with a single target word. To poison an utterance, a source word is chosen at random, forced alignment locates its time span $\Omega$, the trigger is overlaid on that span under an SNR $\geq 22$ dB constraint, and only that one word is swapped in the transcript. At inference the same trigger placed over any spoken word is meant to force that position to decode as the target word, which gives the attacker localized "semantic flips" (e.g. denied $\rightarrow$ allowed) and lets multiple codebooks be composed into sentence-level edits.

The empirical study is fairly broad: two languages (Common Voice v23 English, v24 Lithuanian), four backbones spanning different decoder families (Whisper-Small/Medium, MMS, SpeechT5), and three model sizes. The headline result is an average attack success rate of $89.4\%$, with the attack surviving the two preprocessing defenses (transcript-frequency filtering and VAD) that reduce the BadNet/Blended baselines to near zero. The authors then adapt four optimization-based classification defenses (ABL, ANP, SAU, I-BAU) to ASR and report that suppressing GhostWord drives attack success from $89.4\%$ down to $28.3\%$ but pushes clean WER from $21.9$ to $47.2$. Section 5 offers a theoretical argument -- via a linear-softmax model and a $O(\sqrt{\log K})$ bound on pairwise weight similarity -- that this robustness--accuracy trade-off becomes unavoidable as the number of output classes $K$ grows, which is meant to explain why large-vocabulary ASR is especially hard to defend. Ablations cover poison rate (down to $1\%$), codebook size, words per
sentence, model size, and robustness to forced-alignment timing errors.

Strengths I'd highlight: the core reframing from phrase-level to word-level localized triggers is a clean and well-motivated idea, the breadth across languages/architectures is above what's usual for this subfield, and the demonstration that preprocessing defenses kill the baselines but not GhostWord is convincing. The main weaknesses, expanded in the boxes below, are an attack-success metric that is not comparable to the baselines, a theory that is only loosely tied to the ASR setting, and the absence of a perceptual study and of an adaptive/trigger-reconstruction defense.

**Additional Comments:**

To be clear, I think the core idea is good and the preprocessing-defense result is the real strength of the paper -- my recommendation is essentially "major revision" rather than rejection. Most of my concerns are about calibrating the claims to the evidence (the comparison metric, the "inherent trade-off" framing, the theory's scope) and filling two evidence gaps (a perceptual study and an adaptive defense). If those are addressed I'd be inclined to support acceptance under TMLR's criteria. The localized "semantic flip" framing and the codebook composition are nice, and I'd encourage the authors to lean on those qualitative contributions rather than on a single headline percentage.

**Audience:**

Yes

**Audience Explanation:**

ASR security is an active area, and the move from phrase-level to word-level, time-localized poisoning is the kind of concrete, reusable idea that researchers working on audio backdoors and on backdoor defenses will want to be aware of. The finding that the cheap preprocessing defenses which neutralize prior ASR attacks do not touch a localized word-level trigger is genuinely useful, because it tells the defense community that "filter the repeated transcripts / run VAD" is not a general answer. The breadth of the evaluation (two languages including a low-resource one, four architecturally distinct backbones, multiple model sizes) also makes the empirical observations more broadly informative than a single-model study would be.

Even readers who end up skeptical of the theoretical section or of the headline success metric will find the empirical setup, the codebook formulation, and the defense-adaptation effort worth reading. This sits squarely within the interests of a meaningful slice of the TMLR audience.

**Broader Impact Concerns:**

This is an offensive contribution: a new, harder-to-detect backdoor attack against ASR systems that the paper itself repeatedly frames as safety-critical (in-vehicle interfaces, voice banking, smart environments). I could not find any Broader Impact / ethics statement in the submission, and the authors state that code and defense implementations "will be publicly released." Given the dual-use nature of the work, I think a Broader Impact Statement is required before publication, and it should cover at least the following: (i) an explicit dual-use discussion and the authors' rationale for release; (ii) whether any responsible-disclosure steps were taken with maintainers of the models /
datasets involved; (iii) concrete defensive guidance or candidate mitigations the authors recommend, rather than leaving the contribution one-sidedly offensive; and (iv) confirmation that all experiments used public data and did not target any deployed production system. I'd consider adding this statement a condition for acceptance, not optional.

**Claims And Evidence:**

No

**Claims Explanation:**

I want to be clear up front that I think the central observation is real: the experiments in Tables 1 and 7 do convincingly show that transcript-frequency filtering and VAD collapse the BadNet/Blended baselines to roughly zero while leaving GhostWord essentially intact. That specific claim is well supported. My "No" is about several of the broader, headline-level claims, which I found overstated relative to the evidence.

First, the flagship number is not measuring the same thing as the baselines it's printed next to. $\text{ASR}_{\text{attack}}$ for GhostWord is defined as an exact match of a single target word at the insertion position, whereas for BadNet/Blended it is an exact match of the entire fixed target sentence (Sec. 6 / Appendix J). Producing one correct token is a much easier bar than producing a whole sentence, so reporting "$89.4\%$ vs. baselines" in the abstract and Table 1 reads as a like-for-like comparison when it isn't. The authors do acknowledge this in Appendix M, but the budget-fairness discussion is buried and the main-text framing still invites the wrong reading.

Second, the claim that backdoor mitigation defenses inherently fail (only working at a prohibitive clean-WER cost) is confounded. All four defenses are classification methods retargeted to ASR, and some of the reported runs look less like a graceful trade-off and more like the defense simply breaking the model -- e.g. ABL on MMS/Lithuanian gives clean WER $\approx 100.6$ in Table 7, which is a collapsed model, not evidence of an unavoidable trade-off. More importantly, the most natural defense for this attack is never tried: each codebook's trigger is a single fixed noise pattern reused across all of its poisoned samples, which makes it a candidate for cross-sample correlation or trigger-reconstruction (Neural-Cleanse / STRIP-style) detection. Without testing an adaptive or trigger-synthesis defense, "resilient to all evaluated defenses" is too strong.

Third, I wasn't convinced the theory supports the verbal claim that the trade-off is "unavoidable." Section 5 reduces everything to a linear classifier over class-conditional Gaussians with a single gradient-ascent step, and asserts that other defenses "can be reduced to this setting" without showing it. It ignores the sequence-to-sequence / CTC structure, the autoregressive decoding, the implicit LM prior, and the fact that real token-embedding geometry is highly correlated rather than the i.i.d. subgaussian weights the $O(\sqrt{\log K})$ argument assumes. Figure 2 is a hand-drawn toy, and the analysis never connects quantitatively (even in order of magnitude) to the observed
$21.9 \rightarrow 47.2$ WER jump. As written it's a useful intuition, not a justification.

Fourth, imperceptibility is asserted repeatedly ("perceptually unchanged", "imperceptible to human listeners") but the only evidence is an SNR $\geq 22$ dB threshold. For an attack whose whole selling point is stealth, I'd expect at least a small listening test (MOS/ABX) or an objective perceptual metric (PESQ/STOI); Appendix L even lists improving perceptual quality as future work, which reads like a concession that the current trigger may be audible.

None of these are necessarily fatal, and I think most are fixable in revision -- but as the paper currently stands, several of the claims a reader takes away from the abstract are not backed by comparable, convincing evidence.

**Requested Changes:**

I've marked each item as [Critical] (needed for me to move toward acceptance) or [Strengthen] (would improve the paper but isn't blocking).

1. [Critical] Make the attack-success comparison apples-to-apples. Either report GhostWord under a sentence-level success criterion, or report BadNet/Blended under a per-target-token criterion, so that the numbers placed side by side in Table 1 use a common definition of success. At minimum, move the fairness/budget discussion from Appendix M into the main text and stop juxtaposing the $89.4\%$ figure with baseline numbers that mean something different.

2. [Critical] Evaluate at least one adaptive or trigger-reconstruction defense. Given that each codebook trigger is a fixed, reused additive-noise pattern, a cross-sample correlation / matched-filter or Neural-Cleanse / STRIP-style detector is the obvious thing to try. The "resists all evaluated defenses" claim needs this to be credible.

3. [Critical] Address the collapsed defense runs. For cases like ABL on MMS/Lithuanian where clean WER reaches $\sim 100$, please indicate whether the model has degenerated, and either re-tune those runs or exclude them from the "inherent trade-off" narrative. As-is they weaken rather than support the argument.

4. [Critical] Add perceptual validation of imperceptibility. A small human listening study (MOS or ABX) and/or an objective metric (PESQ, STOI) on triggered vs. clean audio. SNR alone does not establish that the trigger is inaudible.

5. [Strengthen] Reframe Section 5 honestly. State the assumptions of the linear-softmax reduction explicitly, justify (or drop) the claim that the other defenses reduce to single-step gradient ascent, and acknowledge that the i.i.d. subgaussian model ignores correlated token geometry. Presenting it as motivating intuition rather than a guarantee would be more defensible. If you can tie the $\sqrt{\log K}$ effect even loosely to the measured WER increase, that would help a lot.

6. [Strengthen] Report variance. All tables are single-run point estimates given to $0.1\%$. Please add at least 2--3 seeds and error bars for the key comparisons (no-defense vs. best defense per model), so borderline differences (e.g. ANP vs. I-BAU) are interpretable.

7. [Strengthen] Be upfront about scalability and the inference-time threat model. The $89.4\%$ headline uses a 4-word codebook; ASR drops to $\sim 60$--$69\%$ at 10 codebooks (Table 3), so the abstract should not imply broad-vocabulary efficacy. Separately, the inference-time requirement that the attacker forced-align and overlay a trigger on the victim's audio is a strong capability assumption and deserves more than the brief mention in Limitations -- ideally an experiment on noisier / more spontaneous speech rather than only clean read speech from Common Voice.

8. [Strengthen] Fix internal inconsistencies and typos. SAU is "Shared Adversarial Unlearning" in Sec. 4.4.1 but appears as "Smoothing-based Attention Uncertainty" near Table 10 in Appendix H.2 -- one is wrong. Also "Defence Machanisms" and "Implementaion" need a copy-edit pass.

---

> ### Author Response · Authors · 2026-07-20
>
> We appreciate the reviewer's constructive feedback. Every point raised has been thoroughly addressed in the updated manuscript. Below, we provide a summary of our revisions and indicate where the corresponding changes can be found in the revised text (highlighted in blue).
>
> # R1
>
> Thank you for this helpful suggestion. We have revised the evaluation protocol to ensure an apples-to-apples comparison across all attacks. Specifically, we replaced the previous sentence-level attack success metric used for BadNet and Blended with a unified target-word-level attack success rate, which measures the fraction of target words correctly generated at their designated positions. Since GhostWord targets a single word per utterance, its evaluation naturally follows the same criterion (Appendix Section J.2 has been updated accordingly). Therefore, all ASRattack values reported in Tables 1 and 10 are now computed under the same definition of attack success, enabling a direct and fair comparison among GhostWord, BadNet, and Blended.
>
> Following the reviewer’s suggestion, we have also moved the fairness and budget discussion from Appendix M to the main manuscript (Section 7.6) to make the comparison assumptions and resource considerations clearer.
>
> **Corresponding updates in the revision: Experiments (Line 7), Section 7.6, Appendix Section J.2, and Tables 1 and 10.**
>
> # R2
>
> We appreciate the reviewer’s comment. As this concern was shared by multiple reviewers, we address it in detail in Common Response 4 (CP-4), taking into account suggestions from all reviewers. We kindly refer the reviewer to that section for further discussion.
>
> # R3
> We thank the reviewer for this sharp observation. We agree that the collapsed clean WER is a sign of model degeneration, which is not aligned with our intended trade-off justification. After reviewing all reported results, we found that the ABL setting on MMS/Lithuanian is the only case exhibiting a severe clean WER collapse. This was due to an implementation mistake in this specific experiment: the unlearning phase of ABL used an unlearning learning rate that was substantially larger than the value selected through our validation-based hyperparameter study. Using such a high learning rate led to unstable optimization and model collapse. We have corrected this issue by reverting to our selected learning rate. We reran the affected experiment under this corrected configuration, and the results are now consistent with the expected behavior. The updated results are reported in Table 10.
>
> # R4
>
> We appreciate the reviewer’s comment. Since this issue was also noted by other reviewers, we address it comprehensively in Common Response 3 (CP-3) and kindly direct the reviewer to that discussion.
>
> # R5
>
> We appreciate the reviewer’s comment. Since this concern was also raised by multiple reviewers, we provide a detailed response in Common Response 1 (CP-1), incorporating feedback and suggestions from all reviewers. We kindly refer the reviewer to that section for further details.
>
> # R6
>
> As this concern was also raised by multiple reviewers, we have repeated the main experiments in Table 1 with three different random seeds and now report the mean performance together with the standard deviation. We have updated Table 1 accordingly to include the associated uncertainty, resulting in a more robust and reliable evaluation of our results.
>
> # R7
>
> We thank the reviewer for this valuable suggestion. Regarding the first point, we agree that the broad-vocabulary efficacy of our approach should not be overstated. However, after carefully reviewing the manuscript, we could not identify any statements in the abstract that explicitly claim broad-vocabulary performance. We kindly refer the reviewer to Appendix Section M, which discusses the scalability of GhostWord to larger codebooks, and Section 7.6, which provides a fairness analysis explaining the trade-off between attack stealthiness and effectiveness.
>
> Regarding the second point concerning the inference-time threat model, we believe this concern overlaps with Common Response 2 (CP-2), where we discuss the over-the-air experiment setup in detail. We kindly refer the reviewer to that section for further discussion.
>
> # R8
>
> We thank the reviewer for pointing out these inconsistencies. SAU stands for Shared Adversarial Unlearning, and we have corrected the corresponding mismatch in the Appendix. We also performed a thorough copy-edit of the “Defense Mechanisms” and “Implementation” sections to fix typographical issues. These changes are purely linguistic and do not affect the technical content of the paper, and therefore are not highlighted in blue.

---

### Review · Reviewer_Aom9 · 2026-06-21

**Summary Of Contributions:**

The paper proposes GhostWord, a fine-grained word-level backdoor attack against Automatic Speech Recognition (ASR) systems. Specifically, the attack injects short, temporally localized acoustic triggers and relabels only the corresponding source word to a target word through a codebook-based mechanism. Compared with prior phrase-level many-to-one attacks, this design reduces conspicuous statistical artifacts and enables more precise semantic manipulation while maintaining robustness against simple preprocessing defenses. Extensive experiments demonstrate high attack success rates across multiple ASR architectures and languages. The authors further adapt several optimization-based defense strategies to the ASR setting and provide a theoretical analysis of the observed “robustness–utility” trade-off, showing that mitigating the backdoor often incurs a substantial degradation in clean recognition performance. However, there are still several methodological and experimental issues remaining. Detailed comments are as follows.

**Audience:**

Yes

**Audience Explanation:**

The paper provides a meaningful shift from phrase-level, many-to-one backdoors to word-level, time-localized codebook triggers. This constitutes a clear conceptual advance for generative ASR systems, enabling fine-grained semantic manipulation while avoiding the repeated-target artifacts commonly observed in prior approaches.

**Claims And Evidence:**

Yes

**Claims Explanation:**

1. The proposed multi-trigger, multi-target codebook design supports compositional control over attack behavior at inference time, providing a level of flexibility and expressiveness that is largely absent from existing ASR backdoor attacks.
2. Comprehensive empirical evaluations are provided, covering multiple languages (Common Voice English v23 and Lithuanian v24) and four widely used ASR architectures (Whisper Small, Whisper Medium, MMS, and SpeechT5). The results consistently demonstrate high attack success rates, providing strong evidence of the effectiveness and generalizability of GhostWord.

**Requested Changes:**

1.	The practical applicability of the method is questionable. Specifically, the training depends on accurate forced alignment, and the inference also requires this alignment to place triggers. This raises questions about practicality in the wild and under noisy or spontaneous speech. Moreover, the evaluation is fully digital. No over-the-air tests or runtime alignment/injection strategies are shown, leaving practical feasibility in deployed pipelines an open question.
2.	The perceptual characterization of the triggers is insufficiently justified. Specifically, the proposed triggers consist of random-noise snippets injected at a fixed SNR threshold of 22 dB, but the claim of imperceptibility is not adequately supported, as the paper provides neither perceptual listening tests nor human-subject evaluations. To strengthen the empirical evidence, the authors should include ablation studies on key trigger parameters, such as SNR and duration, and conduct a lightweight perceptual study to quantify trigger detectability and the resulting impact on user-perceived audio quality.
3.	The theoretical justification for the observed “robustness-utility” trade-off remains limited. The analysis is based on a simplified linear-classifier setting and geometric properties of the softmax function, which provides useful intuition but falls short of establishing a formal inevitability result for sequence-to-sequence ASR models. Consequently, it remains unclear whether the reported trade-off reflects a fundamental property of the problem or an artifact of the specific defense mechanisms considered. Moreover, the analysis is restricted to optimization-based defenses, such as unlearning and pruning, and does not examine whether alternative defense paradigms exhibit similar trade-off behavior.
4.	More curiosity about the practicality of the attack under realistic deployment conditions. In particular, how would an attacker achieve reliable real-time localization for trigger insertion? Do you have results for over-the-air or streaming scenarios where alignment is imperfect, and latency constraints apply?

---

> ### Author Response · Authors · 2026-07-20
>
> We thank the reviewer for the feedback. We have addressed all of the reviewer's concerns in the revised manuscript. Below, we summarize our responses and refer to the corresponding changes in the revised draft (highlighted in blue).
>
> # R1 & R4: Over-the-Air and Streaming Practicality
>
>
> ## 1. Over-the-Air (OTA) Channel Evaluation
>
> Thank you for raising this point. As this concern was shared by multiple reviewers, we have addressed it in Common Response 2 (CP-2), we kindly ask reviewer to refer that section.
>
>
> ## 2. Streaming Threat Models and Alignment Practicality
>
> ### Practical Deployment.
>
> ASR backdoor attacks can be categorized into two scenarios: asynchronous (offline) and streaming (online). In the asynchronous setting, the attacker has access to the full audio before inference, enabling precise trigger insertion. In the streaming setting, audio is received continuously, and trigger injection can be performed either digitally (with low-latency processing) or physically over-the-air via a loudspeaker. Our main evaluation follows the asynchronous digital setting, and we additionally demonstrate effectiveness under over-the-air conditions in Table 19.
>
> In streaming over-the-air settings, precise trigger placement is more challenging since the attacker does not know in advance when the target phrase occurs. This limitation is shared with existing targeted ASR backdoor attacks due to the unpredictability of spontaneous human speech, so attacks typically rely on approximate timing, hoping that the trigger overlaps with the desired context.
>
> A more practical streaming setup uses low-latency digital trigger injection, where the audio stream is continuously segmented into short VAD-based chunks (e.g., 2–10 seconds). While listening to the stream, each chunk is processed independently, and triggers are inserted before ASR inference, enabling effective localization while satisfying latency constraints in a streaming setting. In the next part, we evaluate GhostWord’s effectiveness in the discussed streaming threat model.
>
> ### GhostWord  under streaming thread model with digital trigger injection
>
> We evaluate GhostWord in the streaming setting described in above. The audio stream consists of 50 sentences recorded continuously in an indoor environment (Whisper and MMS models). The stream is segmented online using a VAD module into ~3s chunks, and triggers are inserted digitally before ASR inference for each chunk, enabling low-latency incremental processing.
>
> As shown in Table below, GhostWord maintains a high attack success rate under this streaming pipeline, demonstrating effective trigger localization without access to the full recording and confirming applicability to streaming scenarios.
>
> Table: GhostWord performance on a simulated streaming setting with 50 English sentences.
> |Model|WERclean|WERpoison|ASRattack|
> |-|-|-|-|
> |MMS|20.3|32.5|84.1|
> |Whisper Medium|12.1|23.3|90.0|
>
> ### GhostWord Robustness to Temporal Alignment Errors
>
> Based on Table 11 in the original draft (Table 14 in the revised manuscript), GhostWord remains highly robust to temporal alignment shifts up to the midpoint boundaries between neighboring words.
>
> **Corresponding update in revision: Appendix Section N, Table 19 and 20.**
>
> # R2
>
> We thank the reviewer for this insightful comment. In the revised manuscript, we have added quantitative ablations and a human perceptual study to address this concern.
>
> ## 1. SNR Ablation Study
>
> To better characterize the trade-off between stealthiness and effectiveness, we conduct a systematic ablation over different SNR levels. Results (Table 15) show that GhostWord remains highly effective even under lower SNR settings, indicating robustness under more subtle perturbations.
>
> **Corresponding update in revision: Appendix Sections H.4, Tables 15**
>
> ## 2. Perceptual Evaluation
>
> We appreciate the reviewer for highlighting this point. As this concern was shared by multiple reviewers, we provide a detailed response in Common Response 3 (CP-3) and kindly invite the reviewer to refer to that section.
>
>
> # R3
>
> Thank you for raising this point. As this concern was shared by multiple reviewers, we have addressed it in Common Response 1 (CP-1), we kindly ask reviewer to refer that section.

---

### Review · Reviewer_aV5y · 2026-07-07

**Summary Of Contributions:**

This paper proposes GhostWord. Unlike prior speech backdoor attacks, GhostWord uses a codebook that binds short acoustic triggers to target words. During training, the trigger is inserted into the time span corresponding to a chosen source word in the audio, while only that word is replaced in the transcript. The paper argues that this word-level attack is stealthier than traditional phrase-level backdoors.

The authors evaluate GhostWord on Common Voice English and Lithuanian using multiple ASR backbones, including Whisper-Small/Medium, MMS, and SpeechT5. The main result is that GhostWord achieves an average attack success rate of about 89.4%. They also test several stronger optimization-based backdoor defenses, such as ABL, ANP, SAU, and I-BAU. These defenses can reduce the attack success rate, for example from 89.4% to 28.3%, but at the cost of increasing clean WER from 21.9% to 47.2%, showing a clear robustness–accuracy trade-off.

**Audience:**

Yes

**Audience Explanation:**

Yes. The paper studies a fine-grained ASR backdoor attack that is relevant to researchers interested in speech recognition, trustworthy ML, and model security. The robustness–accuracy trade-off is also likely to be interesting to researchers working on backdoor defenses and reliable ASR systems.

**Broader Impact Concerns:**

I do not have any concerns about the ethical implications of this work.

**Claims And Evidence:**

Yes

**Claims Explanation:**

Yes. The paper proposes the GhostWord attack, which differs from prior phrase-level ASR backdoors by using word-level, time-localized acoustic triggers that can selectively change individual transcribed words. The paper evaluates the attack across multiple ASR backbones and languages, and studies the limitations of several optimization-based defenses, highlighting a robustness–accuracy trade-off.

**Requested Changes:**

1. The threat model should be better justified. GhostWord requires forced alignment to locate the time span of a specific word during training, and also requires reasonably precise trigger placement at the target word position during inference. This is a relatively strong assumption. The authors should add experiments under temporal misalignment, noisy audio, or real over-the-air playback settings.

2. The defense evaluation should report the full robustness–accuracy trade-off for each defense. In particular, the authors should provide ASRattack vs. clean WER Pareto curves and clearly explain how the defense hyperparameters are selected.

3. The stealthiness claim should be supported by more direct evidence. The paper argues that word-level attacks are stealthier than phrase-level attacks, but this could be further validated through human listening tests, audio quality metrics, or other perceptual/statistical detectability analyses.

---

> ### Author Response · Authors · 2026-07-20
>
> We thank the reviewer for their constructive feedback. We have thoroughly addressed every point raised in the updated manuscript. Below, we summarize our key revisions and point to the corresponding changes, which are highlighted in blue in the revised text.
>
> # R1
>
> Thanks for the constructive feedback. Regarding temporal misalignment, we provide an analysis in Section H.3 and Table 14 regarding robustness to forced-alignment temporal misalignment errors. GhostWord remains highly robust to temporal alignment shifts up to the midpoint boundaries between neighboring words.
>
> Based on your thoughtful suggestion, which was shared by other reviewers, we have conducted over-the-air experiments. Since this concern was also raised by other reviewers, we provide a detailed discussion in Common Response 2 (CR-2), incorporating feedback from all reviewers. We kindly refer the reviewer to that section for further details.
>
> # R2
>
> We appreciate the reviewer highlighting this point. As several reviewers raised a similar concern, we have addressed this comprehensively in Common Response 5 (CR-5). We kindly invite the reviewer to consult CR-5 for our full evaluation and discussion.
>
> # R3
>
> We thank the reviewer for highlighting this point. As this concern was shared by multiple reviewers, we provide a detailed response in Common Response 3 (CR-3) and kindly invite the reviewer to refer to that section.

---

### Review · Reviewer_BMdS · 2026-07-11

**Summary Of Contributions:**

This paper proposes GhostWord, a dirty-label poisoning attack against automatic speech recognition systems. Rather than mapping one acoustic trigger to a fixed sentence, GhostWord constructs a codebook of approximately 400 ms acoustic triggers, each associated with a target word. Poisoned examples overlay a trigger on the forced-aligned span of a source word and replace that word in the transcript. The resulting model can perform localized semantic substitutions and compose several target words. Experiments cover Common Voice English and Lithuanian with Whisper-Small, Whisper-Medium, MMS, and SpeechT5. The attack is effective across these settings, and the paper includes ablations on poisoning rate, codebook size, trigger density, temporal misalignment, and data augmentation. The fine-grained attack formulation is interesting and the empirical breadth is a strength. However, the current evidence does not support the stronger claims concerning stealth, practical transferability, resistance to defenses, or an inevitable clean-accuracy trade-off. The attack-success metric and experimental protocol also require important clarification and additional controls.

**Additional Comments:**

The paper is generally readable, and the codebook formulation and contextual-WER analysis are useful. The most important revision is to separate what the experiments currently establish—effectiveness under controlled digital poisoning—from the stronger claims of imperceptibility, practical transfer, broad defense resistance, and theoretically unavoidable degradation.

**Audience:**

Yes

**Audience Explanation:**

Fine-grained, composable word-level poisoning is a relevant and potentially important threat for ASR. It exposes limitations of evaluations centered on fixed sentence-level targets and raises useful questions about how sequence structure changes backdoor detection and mitigation. Researchers in trustworthy ML, speech processing, dataset security, and backdoor defense would likely find the attack formulation and cross-architecture results interesting. The work could make a meaningful contribution if the authors validate the attack under more realistic conditions and narrow or substantiate the theoretical and transferability claims.

**Broader Impact Concerns:**

The work has substantial dual-use potential because it develops a more precise mechanism for covertly changing safety-sensitive speech commands and transcripts. The paper should include a dedicated discussion of misuse scenarios, attacker requirements, affected deployment settings, disclosure considerations, and safeguards for releasing poisoned datasets, trigger codebooks, trained models, and attack-generation code. The current limitations section does not adequately address these risks.

**Claims And Evidence:**

No

**Claims Explanation:**

Evidence Assessment

The experiments convincingly establish that the proposed poisoning procedure can produce high measured attack success under the authors’ controlled fine-tuning setup. Results are reported across several architectures and two languages, and the poison-rate, codebook-size, temporal-shift, contextual-WER, and augmentation ablations are useful.

Several issues prevent the evidence from supporting the paper’s broader conclusions.

First, the attack-success metric is underspecified and internally inconsistent. Appendix J initially describes success as the target word appearing anywhere in the decoded output, but Equation 15 evaluates the word at the “trigger insertion location.” An ASR decoder does not naturally provide a one-to-one mapping between waveform time and decoded word/token position, particularly for subword tokenization and insertion/deletion errors. The paper must specify the alignment and normalization algorithm and demonstrate that results are not inflated by naturally occurring target words. There is no reported false-activation rate on an unpoisoned model, random-trigger control, mismatched-codebook control, or per-target-word result.

Second, stealth is asserted rather than demonstrated. A 22 dB segment-level SNR does not establish perceptual imperceptibility, especially for a fixed 400 ms mid-band noise waveform inserted into speech. No listening study, perceptual metric, spectral detectability test, or audio-text consistency detector is evaluated. Replacing the transcript while preserving the original spoken word also creates a localized audio-label mismatch that forced alignment or model-based dataset auditing may detect. Testing only VAD and exact repeated-transcript filtering does not justify the claim that GhostWord avoids simple preprocessing or dataset-inspection defenses.

Third, the defense evaluation does not isolate a fundamental robustness–accuracy trade-off. The classification defenses are substantially modified for sequence generation, with model-specific tuning and single reported operating points. Some adaptations appear intrinsically destructive—for example, treating any sequence deviation as adversarial success and comparing only overlapping output prefixes. The paper reports neither defense Pareto curves nor corresponding runs on clean, non-backdoored models. It is therefore unclear how much WER degradation is specific to GhostWord rather than to an aggressive or poorly calibrated ASR adaptation. The phrase-level baselines are also unusually easy to detect because they repeatedly assign one complete target transcript. The reproduced ASR-specific baselines lack official implementations and are evaluated narrowly.

Fourth, the theoretical section does not establish the claimed inevitability result. It analyzes one gradient-ascent step for a linear K-class softmax classifier, not CTC or autoregressive ASR, and does not show that ANP, SAU, I-BAU, and ABL reduce to this setting. Equation 12 omits the dependence on feature dimension and vector normalization needed for the claimed maximum-inner-product scaling. Moreover, when the target probability fj is near one, Equation 11 implies that the aggregate non-target update is proportional to 1-fj and is therefore small; the subsequent claim of necessarily large destructive updates does not follow. No bound on clean error is derived, and the assumptions of independent class vectors and Gaussian, linearly separable classes are not connected to learned ASR token representations. At most, this section provides an intuition for one unlearning method, not proof of an unavoidable high-vocabulary trade-off.

Finally, the empirical protocol lacks uncertainty estimates and contains reproducibility inconsistencies. The main text states four epochs with a fixed learning rate, whereas Appendix K specifies ten epochs and model-dependent learning rates. Seeds, variance, codebook words, trigger-specific performance, selection criteria for the reported defense points, and the exact aggregation producing the headline averages are missing. Testing only small, low-resource subsets of one dataset family also limits the generality of the conclusions.

**Requested Changes:**

1. (Critical) Correct or substantially revise the theoretical contribution. State a formal theorem with complete assumptions and a valid clean-error result, including feature dimension and normalization, or reframe the section explicitly as limited intuition. Do not claim that clean degradation is inevitable or that all evaluated defenses reduce to the analyzed gradient-ascent update without proof.

2. (Critical) Define the attack-success computation precisely for CTC and encoder-decoder models, including word normalization, subword aggregation, temporal/output alignment, and handling of insertions and deletions. Report per-codebook results and controls using unpoisoned models, random triggers, mismatched triggers, and natural target-word occurrence rates.

3. (Critical) Validate stealth and practical feasibility. Include audio examples and human or established perceptual evaluation, quantitative trigger-detection tests, and defenses based on denoising, spectral analysis, audio-text consistency, forced-alignment residuals, and word-frequency changes. Evaluate playback/recording, compression, resampling, and realistic timing uncertainty, or explicitly narrow the threat model to digital access with known word boundaries.

4. (Critical) Rework the defense comparison. Report clean-WER versus attack-success Pareto curves across defense strengths, explain hyperparameter selection without using test results, and run every adapted defense on matched clean models. Include stronger word-level dataset-auditing baselines and avoid treating repeated fixed transcripts as representative of all competing ASR attacks.

5. (Critical) Improve statistical and experimental rigor. Run multiple seeds, report uncertainty, disclose the target/source words and poisoning construction, and provide per-language, per-model, per-word, and per-trigger breakdowns. Resolve the four-versus-ten-epoch contradiction and document how all headline averages are calculated.

6. (Minor) Replace “transferable across languages and models” with “effective across separately trained languages and models” unless the same poisoned corpus or trigger is explicitly evaluated under a genuine cross-model or cross-language transfer protocol.

7. (Minor) Clarify how a fixed-length trigger is handled when the aligned source word is shorter than the trigger, and specify how multiple triggers are placed without overlapping adjacent words.

8. (Minor) Correct presentation issues, including inconsistent attack-success descriptions, “four codebooks” versus “ten words” terminology, table cross-references, typographical errors, and inconsistent model/training descriptions.

---

> ### Author Response · Authors · 2026-07-20
>
> We appreciate the reviewer’s constructive feedback. We have carefully addressed all raised points in the revised manuscript. Below, we summarize the corresponding revisions and indicate their locations in the updated text, with all changes highlighted in blue.
>
> # R1
>
> We appreciate the reviewer’s attention to this aspect of our analysis. To address this shared concern thoroughly, we have consolidated our response and revision details into Common Response 1 (CR-1). We invite the reviewer to consult that section for full details regarding our updates.
>
> # R2
>
>
> We address this concern in three parts.
>
> (1) Natural occurrence of target words. We agree that naturally occurring target words could artificially inflate the attack success rate. To eliminate this issue for GhostWord, we apply a strict vocabulary filter during the construction of the poisoned test set. For each evaluation sample, we only assign a codebook whose target word does not appear in the original clean transcription. If a target word is already present, that codebook is skipped and an alternative codebook is selected. As a result, any successful generation of the target word can be attributed solely to the trigger rather than to its natural occurrence.
>
> (2) Attack-success computation. Our attack success rate is computed at the word level from the final decoded transcription, treating the ASR system as a black box. We do not compare intermediate tokens; therefore, our metric is independent of whether the underlying model is CTC-based or encoder-decoder based. For GhostWord, the designated target position is determined from the reference transcription: the inserted target word is expected to appear between two known neighboring words. Since the target word is guaranteed not to occur naturally in the original transcription (as described above), any occurrence of the target word in the decoded transcription can be attributed to the trigger.
>
>
> (3)  Per-codebook attack success rate for GhostWord.
> We report the attack success rate of GhostWord for each individual codebook to provide a more detailed analysis of attack performance.
>
> | Language | Model Name | ASR$_{\text{codebook1}}$ | ASR$_{\text{codebook2}}$ | ASR$_{\text{codebook3}}$ | ASR$_{\text{codebook4}}$ |
> | :--- | :--- | :--- | :--- | :--- | :--- |
> | | Whisper Small | 89.4 | 97.6 | 86.7 | 88.8 |
> | **English**| SpeechT5 | 91.3 | 90.5 | 87.3 | 84.2 |
> | | MMS | 85.4 | 89.7 | 78.4 | 77.5 |
> | | Whisper Medium | 89.9 | 97.4 | 90.8 | 94.5 |
>
> **Corresponding update in the revision: Appendix Section O**
>
> In addition, following Reviewer cqPp R.1's suggestion, we have revised the evaluation of BadNet and Blended to use the same unified word-level attack success metric, ensuring a fair comparison across all attack types. Appendix J.2 has been updated accordingly.
>
> **Corresponding update in the revision: Appendix Section J.2, Tables 1 and 10.**
>
> # R3
>
> We appreciate the reviewer’s attention to this matter. To address this shared concern thoroughly, we have provided an expanded evaluation and discussion in Common Response 4 (CR-4), incorporating insights from all reviewers. We invite the reviewer to consult that section for full details.
>
> # R4
>
> We appreciate the reviewer’s attention to this matter. Since this concern was also raised by other reviewers, we provide a detailed evaluation and discussion in Common Response 5 (CR-5), incorporating feedback from all reviewers. We kindly refer the reviewer to that section for further details.

---

> > ### Author Response · Authors · 2026-07-20
> >
> > # R5
> >
> > We thank the reviewer for highlighting the importance of improving statistical and experimental rigor. We have addressed these concerns as follows.
> >
> > **Multiple seeds and uncertainty.** We have repeated the main experiments reported in Table 1 using three different random seeds and report the mean performance along with the standard deviation. We have updated Table 1 to include the corresponding uncertainty, providing a more reliable evaluation of our results.
> >
> > **Target/source words and poisoning construction.** To examine whether GhostWord's effectiveness depends on a particular set of target words, we evaluate the attack separately on each of the four codebooks. As shown in Table 21, GhostWord consistently achieves high attack success rates across all codebooks and ASR backbones. Although minor variations exist due to differences in target words and model architectures, no single codebook dominates the overall performance.
> >
> > **Headline average calculation.**  We appreciate the reviewer’s comment regarding the calculation of headline averages. The detailed calculation procedure is provided in Section 6. For further clarification, we have also updated the Contribution paragraph in the Introduction to explicitly describe how the reported averages are computed.
> >
> > **Epoch setting clarification.** Thank you for pointing this out. The statement in the implementation details was incorrect. Unless otherwise specified, all experiments were conducted for 10 epochs. We have corrected this mistake in the revised manuscript to avoid any confusion.
> >
> > **Corresponding updates in the revision: Introduction (Contribution paragraph), Section 6 (Implementation Details, Line 2), Appendix O, and Table 1 and 21.**
> >
> > # R6
> >
> > Thank you for this suggestion. We have updated the contribution paragraph in the Introduction by replacing "transferable across languages and models" with more accurate wording that reflects our experimental setting.
> >
> >
> > # R7
> > This case is already described in Section 4.2 ("Trigger Injection Function"), and we have further clarified the description. Specifically, if the trigger duration exceeds that of the aligned source-word interval, we extend (pad) the interval to accommodate the full trigger while ensuring that it does not overlap with adjacent word segments. Conversely, if the aligned source-word interval is longer than the trigger, we center the trigger within the interval and leave the remaining portions unchanged. With this design, trigger insertions remain localized and do not result in overlap, even in a multiple-trigger setting.
> >
> >
> > # R8
> >
> > Thank you for pointing out these presentation issues. We have thoroughly revised the manuscript to address them, including clarifying the attack-success metric, resolving the "four codebooks" versus "ten words" terminology, correcting table cross-references, fixing typographical errors, and improving the consistency of the model and training descriptions.

---

### Review · Reviewer_H7EN · 2026-07-15

**Summary Of Contributions:**

The authors present GhostWord, a new backdoor attack for ASR systems. Instead of poisoning whole phrases, which is easy to catch, they map short acoustic triggers (~400 ms) to specific target words using a codebook. They inject this trigger exactly where the source word is spoken in the audio and just swap that one word in the transcript. It works really well, hitting an 89.4% success rate across models like Whisper, MMS, and SpeechT5. They also show that trying to defend against this using current unlearning methods just wrecks the model's clean accuracy. Overall, it's a clever approach that exposes a real blind spot in ASR security.

**Additional Comments:**

Really enjoyed reading this. The writing is clear, the figures (especially the pipeline in Figure 1 and the trade-off charts in Figure 3) are super helpful, and the core idea is just very peculiar.

**Audience:**

Yes

**Audience Explanation:**

Anyone working in AI safety, speech processing, or adversarial ML will want to read this. ASR systems are everywhere right now, and showing that they can be easily manipulated at the word level without setting off alarms is a big deal for the security community.

**Claims And Evidence:**

Yes

**Claims Explanation:**

The experiments are incredibly thorough. They tested the attack on multiple models and languages, comparing it directly to existing phrase-level attacks. The data clearly shows that while simple defenses wipe out the old attacks, GhostWord slips right through. They also back up the defense trade-off with a solid mathematical explanation of how unlearning shifts probability mass around in high-vocabulary models. It all holds together really well.

**Requested Changes:**

I honestly don't have any major blockers for acceptance, but a couple of things would make it even better.

- I'd love to see a brief discussion or test on how well the attack holds up in super noisy environments or with very casual, conversational speech, since it relies heavily on forced alignment.

- since you proved that unlearning defenses ruin clean performance, maybe add a sentence or two pointing future researchers toward non-optimization-based defenses that might actually stand a chance.

---

> ### Author Response · Authors · 2026-07-20
>
> We sincerely appreciate the reviewer's constructive feedback. We have carefully addressed all of the raised points in the revised manuscript. Below, we summarize the corresponding revisions and indicate where they can be found in the updated manuscript. All modifications are highlighted in blue.
>
>
> # R1
>
> Thank you for your positive feedback.
>
> Regarding temporal alignment, we already provide an analysis in **Section H.3** and **Table 14**, evaluating the robustness of GhostWord to forced-alignment temporal misalignment errors. Our results show that GhostWord remains highly robust to alignment shifts up to the midpoint boundaries between neighboring words.
>
> Based on your thoughtful suggestion, which was also shared by other reviewers, we further conducted over-the-air experiments to evaluate the attack under more realistic acoustic conditions. We provide a detailed discussion of these experiments in **Common Response 2 (CR-2)**, incorporating feedback from all reviewers. We kindly refer the reviewer to that section for further details.
>
> # R2
>
> Thank you for this insightful suggestion. We have added a Future Work discussion in Section 8 highlighting this point and encouraging research into non-optimization-based defense paradigms.

---

### Review · Reviewer_nUKB · 2026-07-15

**Summary Of Contributions:**

This paper introduces GhostWord, a data poisoning backdoor attack against automatic speech recognition, and its main idea is to move poisoning from the sentence level down to the word level. Prior ASR backdoors tie one fixed trigger to one fixed target transcript, which leaves obvious traces like repeated transcriptions or triggers sitting in non speech regions. GhostWord instead builds codebooks, where each entry pairs a short additive noise trigger of about 400 ms with a single target word. To poison an utterance the attacker picks a source word at random, uses forced alignment to find its time span, overlays the trigger under a 22 dB SNR constraint, and swaps only that one word in the transcript. At inference the same trigger is meant to force whatever position it covers to decode as the paired target word, giving localized semantic flips and composable sentence level edits. Across two Common Voice languages and four backbones (Whisper Small and Medium, MMS, SpeechT5), it reports an average attack success rate of 89.4 percent, shows the cheap preprocessing defenses that flatten the baselines leave GhostWord untouched, and argues that stronger defenses suppress it only at a steep clean WER cost.

I think the core reframing is clean and well motivated, and the finding that transcript filtering and VAD kill the baselines but not GhostWord is convincingly backed by Tables 1 and 7. But my reservations are about whether the bigger claims match the evidence. The headline number is not measured the same way as the baselines beside it, since GhostWord only needs one word to match while the baselines need a whole sentence, so I do not read 89.4 percent as apples to apples. The theory rests on a linear softmax toy model that is never tied to CTC or autoregressive decoding, yet gets presented as an inevitability result. On defenses I was not convinced the trade off is fundamental, since some runs look like the model simply collapsed, there are no Pareto curves, and the most natural adaptive defense against a fixed reused trigger is never tried. Stealth is also asserted from an SNR threshold with no listening test, and the inference time threat model quietly assumes an attacker who can already align and inject into the victim audio.

**Audience:**

Yes

**Audience Explanation:**

ASR security is an active area, and the shift from phrase level to word level, time localized poisoning is the kind of concrete, reusable idea that researchers working on audio backdoors and on backdoor defenses would want to be aware of. I think the most informative finding is that the cheap preprocessing defenses which neutralize prior ASR attacks do not touch a localized word level trigger, since that tells the defense community that filtering repeated transcripts or running VAD is not a general answer. The breadth of the evaluation, covering two languages including a lower resource one and four architecturally distinct backbones, also makes the empirical observations more broadly informative than a single model study would be. Even readers who end up skeptical of the theoretical section or the headline success metric should still find the codebook formulation and the defense adaptation effort worth reading.

**Broader Impact Concerns:**

The submission does not contain a Broader Impact or ethics statement. One is needed because this is a purely offensive contribution that develops a harder to detect way of covertly altering safety sensitive speech commands, in exactly the deployment settings the paper highlights, and it states that code and data will be publicly released. I would ask the authors to add a statement covering the dual use risks and attacker assumptions, a rationale for release together with any responsible disclosure steps, some concrete defensive guidance, and confirmation that all experiments used public data without targeting any deployed system.

**Claims And Evidence:**

No

**Claims Explanation:**

One central claim is well supported. Tables 1 and 7 do convincingly show that transcript filtering and VAD collapse the BadNet and Blended baselines while leaving GhostWord intact, and I think that result is genuinely useful. My "No" reflects three headline claims that seem to me somewhat stronger than the current evidence supports.

1. The flagship success number does not appear to be measured on the same basis as the baselines printed next to it. GhostWord counts an exact match of a single target word at the insertion position, whereas the baselines require an exact match of the whole target sentence. Since one correct token is an easier bar than a full sentence, placing 89.4 percent beside the baseline numbers can read as a like for like comparison when it may not be one. The fairness discussion in Appendix M helps, but it is somewhat buried, and the main text framing could still invite the wrong reading.

2. The claim that mitigation defenses inherently fail unless clean WER is wrecked seems to me partly confounded. The defenses are classification methods retargeted to ASR with model specific tuning and a single operating point each, and a few runs look less like a graceful trade off than like the defense simply breaking the model, for example ABL on MMS Lithuanian with clean WER near 100 in Table 7. I did not find Pareto curves or matched runs on clean models. It might also be worth noting that each codebook trigger is a fixed noise pattern reused across all its poisoned samples, which could make it a candidate for cross sample correlation or trigger reconstruction detection, and no adaptive defense of this kind seems to be tried, so resilient to all evaluated defenses may be a bit strong.

3. I was not fully convinced the theory or the stealth claim is established. Section 5 reduces things to a linear softmax classifier with a single gradient ascent step and states that the other defenses reduce to this setting without quite showing it, and it sets aside the sequence structure and the correlated token geometry that the O(sqrt(log K)) argument assumes away, so it reads to me more as intuition than as an inevitability result. Separately, imperceptibility is supported mainly by a 22 dB SNR, without a listening study or perceptual metric.

ASR security is an active area, and the shift from phrase level to word level, time localized poisoning is the kind of concrete, reusable idea that researchers working on audio backdoors and on backdoor defenses would want to be aware of. I think the most informative finding is that the cheap preprocessing defenses which neutralize prior ASR attacks do not touch a localized word level trigger, since that tells the defense community that filtering repeated transcripts or running VAD is not a general answer. The breadth of the evaluation, covering two languages including a lower resource one and four architecturally distinct backbones, also makes the empirical observations more broadly informative than a single model study would be. Even readers who end up skeptical of the theoretical section or the headline success metric should still find the codebook formulation and the defense adaptation effort worth reading.

**Requested Changes:**

See above

---

> ### Author Response · Authors · 2026-07-20
>
> We appreciate the reviewer's helpful feedback. All comments have been fully addressed in the revised manuscript. A summary of our responses and the locations of the corresponding edits (highlighted in blue) is provided below.
>
> # R1
>
> Thank you for this helpful suggestion. We have revised the evaluation protocol to ensure an apples-to-apples comparison across all attacks. Specifically, we replaced the previous sentence-level attack success metric used for BadNet and Blended with a unified target-word-level attack success rate, which measures the fraction of target words correctly generated at their designated positions. Since GhostWord targets a single word per utterance, its evaluation naturally follows the same criterion (Appendix Section J.2 has been updated accordingly). Therefore, all ASRattack values reported in Tables 1 and 10 are now computed under the same definition of attack success, enabling a direct and fair comparison among GhostWord, BadNet, and Blended.
>
> Following the reviewer’s suggestion, we have also moved the fairness and budget discussion from Appendix M to the main manuscript (Section 7.6) to make the comparison assumptions and resource considerations clearer.
>
> **Corresponding updates in the revision: Experiments (Line 7), Section 7.6, Appendix Section J.2, and Tables 1 and 10.**
>
> # R2
>
> Thank you for your valuable feedback. We address the comments in three parts.
>
> **First, regarding ABL on MMS Lithuanian.**
> We agree that the collapsed clean WER indicates model degeneration and does not support our intended trade-off analysis. After reviewing all reported results, we found that the ABL experiment on MMS/Lithuanian was the only case exhibiting such a severe collapse in clean performance. This issue was caused by an implementation mistake specific to this experiment: during the unlearning phase of ABL, we inadvertently used an unlearning learning rate that was substantially higher than the value selected through our validation-based hyperparameter tuning. This excessively high learning rate resulted in unstable optimization and model collapse. We have corrected this issue by restoring the validated learning rate and reran the affected experiment. The updated results are now consistent with the expected behavior and are reported in **Table 10**.
>
> **Second, regarding the Pareto curves.**
> We appreciate the reviewer's attention to this issue. Since the same concern was raised by multiple reviewers, we provide a detailed evaluation and discussion in **Common Response 5 (CR-5)**, incorporating feedback from all reviewers. We kindly refer the reviewer to that section for further details.
>
> **Third, regarding adaptive defenses.**
> We appreciate the reviewer's comment. As this concern was also raised by multiple reviewers, we address it in detail in **Common Response 4 (CR-4)**, incorporating feedback and suggestions from all reviewers. We kindly refer the reviewer to that section for a more comprehensive discussion.
>
>
> # R3
>
> Thank you for your valuable feedback. We address the comments in two parts.
>
> **First, regarding the theoretical analysis.**
> We appreciate the reviewer for raising this important point. As this concern was also shared by multiple reviewers, we address it in detail in **Common Response 1 (CR-1)**. We kindly refer the reviewer to that section for a comprehensive discussion.
>
> **Second, regarding the perceptual evaluation.**
> We appreciate the reviewer's comment. Since this concern was also raised by multiple reviewers, we address it comprehensively in **Common Response 3 (CR-3)**, which includes additional perceptual evaluation and discussion. We kindly refer the reviewer to that section for further details.

---

### Author Response · Authors · 2026-07-20
**Common Response 1, 2**

## CR-1: Clarification of the Theoretical Analysis

We sincerely thank the reviewers for their constructive feedback regarding the theoretical analysis in Section 5. In response, we have substantially revised the presentation of the analysis to better position it and clarify its scope and intent.

Specifically, we have revised Section 5 to explicitly present the analysis as an informal theoretical treatment intended to provide a motivating theoretical intuition for the empirically observed robustness–utility trade-off, rather than a formal inevitability theorem or theoretical guarantee for sequence-to-sequence ASR models. To this end, we have implemented the following structural updates:

* **Explicitly reframe the analysis:** The beginning of Section 5 now clearly states that the analysis is intended to provide mechanistic intuition rather than a universal theoretical result or proof of inevitability.
* **Introduce a dedicated Assumptions subsection:** We explicitly describe the simplifying assumptions underlying the analysis, including the linear-softmax approximation, a single gradient-update step, and the i.i.d. sub-Gaussian weight model.
* **Discuss the limitations of the assumptions:** We introduce a new remark that explicitly acknowledges that real ASR vocabularies exhibit correlated token geometry (e.g., phonetically and semantically related tokens). We clarify that the i.i.d. weight model serves as a conservative statistical simplification rather than a realistic description of token embeddings.
* **Narrow the scope of the claims regarding defense mechanisms:** We clarify that the analysis provides a mechanistic explanation directly applying to gradient-ascent and fine-tuning-based updates (optimization-based parameter-update defenses). We no longer imply or suggest that all defense paradigms reduce to a single gradient-ascent step, explicitly noting that detection- and pruning-based defenses involve entirely different mechanisms and are therefore outside the scope of this reduction.

**Corresponding update in revision: Section 5**

## CR-2: Over-the-Air (OTA) Channel Evaluation
To bridge the gap between digital and real-world deployment, we conducted new physical over-the-air experiments using a loudspeaker–microphone setup in a standard indoor acoustic environment (small room, ~1.5m distance, typical background noise). Unlike the digital setting, the trigger undergoes realistic acoustic propagation, including environmental noise and microphone recording artifacts before reaching the ASR model.

Despite these distortions, **GhostWord** maintains a high attack success rate, demonstrating that the learned trigger remains effective under realistic acoustic conditions rather than only in idealized digital evaluation. The results are shown below:


Table 16: GhostWord performance on the OTA setting with 50 English sentences.
|Model|WERclean|WERpoison|ASRattack|
|-|-|-|-|
|MMS|19.8|31.6|82.5|
|Whisper Medium|11.3|21.8|89.2|

**Corresponding update in revision: Appendix Section N.1, Table 19.**

---

### Author Response · Authors · 2026-07-20
**Common Response 3**

## CR3: Perceptual Evaluation

We sincerely thank the reviewers for this valuable suggestion. To directly evaluate the perceptual imperceptibility and stealthiness of GhostWord, we have expanded our evaluation to include both subjective human listening studies and objective speech quality metrics.

Because GhostWord targets a localized word-level trigger rather than manipulating an entire utterance, its goal is to embed a perturbation that is difficult to detect while preserving the naturalness and intelligibility of the speech. Accordingly, we evaluate these properties through a two-stage human perceptual study (20 participants, 50 poisoned samples, totaling 1,000 evaluations) and an objective Short-Time Objective Intelligibility (STOI) analysis.

### Human Perceptual Study
To supplement our SNR constraint, we conducted a two-stage human perceptual study (20 participants, 50 poisoned samples) to validate stealthiness.

* Task 1 (Localization): For each audio sample, participants were asked to identify the noisy word from a set of 3 candidates (poisoned word and 2 distractors) or None. The observed correct identification rate of 42.4% is only slightly above chance, confirming that the perturbation lacks clear artifacts and is difficult for listeners to localize within an utterance.

* Task 2 (Intelligibility): In the second task, participants were explicitly informed which word was poisoned and asked to evaluate its clarity. The majority of participants (over 94%) rated it as clear or understandable. This confirms the injection successfully preserves linguistic clarity and phonetic integrity.

The aggregate response distribution across all 1,000 evaluations (20 participants * 50 samples) is summarized below. Option A represents the poisoned word here for readability, but the candidate order was randomized during testing:

| ID | Question | A | B | C | D |
|----|----------|---|---|---|---|
| Q1 | Noise word (A:Poisoned,B:D1,B:D2,D:None) | 42.4% | 17.3% | 14.7% | 25.6% |
| Q2 | Clarity (A:Clear,B:Understand,C:Hard,D:Not) | 68.5% | 26.2% | 4.1% | 1.2% |


### Objective Metrics (STOI)

In addition to the human listening study, we have incorporated an objective perceptual evaluation using the Short-Time Objective Intelligibility (STOI) metric, as suggested by the reviewer. We computed STOI on 50 randomly selected clean–poisoned audio pairs and obtained an average score of 0.98, indicating that the injected trigger causes negligible degradation in speech intelligibility. This objective result complements the human listening study, providing both objective and subjective evidence that GhostWord preserves speech intelligibility while embedding a perceptually stealthy trigger.


**Corresponding update in revision: Appendix Sections H.5 and H.6, and Table 16.**

---

### Author Response · Authors · 2026-07-20
**Common Response 4**

## CR4: Adaptive Defence
We thank the reviewers for highlighting adaptive defenses. We evaluate GhostWord against five defenses—Adaptive Neural Cleanse, Adaptive STRIP, compression/resampling, Forced-Alignment Residuals, and Word Frequency Analysis—assuming the defender knows the attack uses word-level triggers. Across all defenses, GhostWord shows no reliable detection signature or mitigation.

See **Section 4.4.2** for details. A summary is below.

**Adaptive Neural Cleanse.** Neural Cleanse reverse-engineers a minimal perturbation for each class and detects backdoors by identifying outlier perturbation norms (e.g., using Median Absolute Deviation).

For GhostWord, we first train a model on the poisoned dataset. Given a clean audio sample, we localize a randomly selected word $W$, extract its segment $S$, and search for adversarial perturbations on $S$ that transform $W$ into each of 20,000 English words. We then rank candidate targets by the $L_1$ norm of the required perturbations.
If Neural Cleanse were effective, the true target would appear among the top-ranked candidates (e.g., top-4), enabling detection or removal. However, the target word is not ranked in the top-4; it appears only within the top-200 candidates, with the best-case rank being 29th. Across 10 samples and multiple targets, the trigger is not recoverable using this criterion.

**Adaptive STRIP.**
STRIP is an inference-time Trojan detection method that repeatedly perturbs an input and measures the Shannon entropy of the model's predictions. It assumes that a backdoor trigger dominates the prediction, causing triggered inputs to exhibit consistently low entropy despite perturbations.

Since GhostWord embeds a localized word-level trigger, applying STRIP to an entire utterance introduces variability from unrelated words that can obscure the trigger's entropy signature. We therefore evaluate a stronger adaptive variant that assumes the defender knows the attack is word-level. The defender uses forced alignment to obtain word boundaries and applies STRIP per word.

*Experimental Setup.*
We evaluate this defense on 30 poisoned samples (average transcript length ≈10 words). For each candidate word, we generate 1,000 perturbed versions by injecting random Gaussian, Uniform, Laplace, or Student's t noise (with randomly sampled scales) into only that word segment. Each perturbed audio is transcribed using the poisoned model. Because ASR outputs vary in length, we align each perturbed transcription with the clean transcript to recover the word corresponding to the perturbed segment, then compute the Shannon entropy of the resulting word distribution.

*Results.*
|Position|Average Entropy|
|-|-|
|Target (codebook) words|0.92|
|Non-target words|0.99|

The entropy values are nearly identical, indicating that adaptive STRIP cannot distinguish trigger locations from benign words. Even with forced alignment and word-level perturbations, the expected low-entropy signature of a backdoor does not emerge, suggesting that GhostWord remains robust against this adaptive variant of STRIP.

**Resampe/Compression.**
We evaluate GhostWord under MP3 compression (16 kbps) and 16 kHz → 8 kHz → 16 kHz resampling on the existing poisoned test set without retraining. MP3 compression preserves clean ASR (WER clean: 15.2% → 15.0%) while reducing $ASR_{attack}$ from 90.8% to 61.1%, showing partial mitigation. Since we use a simple Gaussian-noise trigger, an adaptive attacker could design a more compression-robust trigger. Resampling reduces $ASR_{attack}$ to 64.9% but also degrades clean ASR (WER clean: 42.6%), suggesting that the reduction is mainly due to overall ASR degradation rather than targeted trigger removal.
||Condition |WER Clean|WER Poison|ASR$_{attack}$|
|-|-|-|-|-|
|| Original (uncompressed)|15.2|26.0|90.8|
|Whisper Small (English)|MP3 16kbps|15.01|27.8|61.1|
||Resample 16→8→16kHz|42.6|45.3|64.9|

**Forced-Alignment Residuals.**
CTC-based forced alignment provides monotonic frame-to-token alignment but does not verify phonetic consistency. We compare the aligned spans of the original word ($\Omega_{clean}$) and the poisoned target word ($\Omega_{poison}$) at the same location across 50 English samples. The mean duration difference is 60.1 ms (SD 12.6 ms) with a temporal IoU of 80.5%, indicating that poisoned words preserve similar alignment positions and durations. Thus, boundary- or duration-based residual detectors are unlikely to identify GhostWord triggers.
|Metric|Mean|SD|
|-|-|-|
|\|Δ duration\|(ms)|60.1|12.6|
|Temporal IoU |0.805|0.134|

**Word Frequency.**  We analyze whether target-word insertion creates detectable lexical anomalies. As shown in Figure 4, the 100 most frequent words exhibit no observable distribution shift after poisoning. The injected targets blend into the natural Zipfian distribution, preserving corpus statistics and preventing reliable detection by frequency-based defenses.

**Corresponding update in revision: Section 4.4.2**

---

### Author Response · Authors · 2026-07-20
**Common Response 5**

## CR5. Pareto Curve

We appreciate the reviewers' valuable feedback. To address all three aspects of your comments, we have comprehensively updated both our defense evaluation and experimental protocol.

**Robustness–accuracy (Pareto) curves.** Rather than reporting only a single operating point for each defense, we now characterize the robustness–accuracy trade-off across defense strengths. Specifically, for optimization-based defenses (SAU, ABL, ANP, and I-BAU), we vary a single hyperparameter that directly controls the defense strength (e.g., the number of unlearning epochs for SAU and the unlearning learning rate for ABL), while keeping all remaining hyperparameters fixed. The resulting trade-off curves for SAU and ABL, shown in Figures 2 and 3 and evaluated on a dedicated held-out validation set, illustrate the relationship between attack suppression and clean recognition performance. We performed the same analysis for ANP and I-BAU and observed the same qualitative robustness–accuracy trade-off. To avoid making the paper unnecessarily dense with multiple additional figures exhibiting the same behavior, we summarize these observations in the main text rather than including the corresponding trade-off plots. Across all optimization-based defenses, increasing the defense strength consistently reduces attack success at the cost of increasing clean WER, with progressively diminishing robustness gains as the defense becomes more aggressive.

**Hyperparameter selection without using the test set.** We have clarified the complete hyperparameter selection procedure in the paper (Section 4 and Appendix K). All defense hyperparameters were selected **exclusively on a dedicated held-out validation set**, which is completely disjoint from both the training and test sets. We first performed the hyperparameter study on the Whisper-Small (English) setting and then **fixed the resulting hyperparameter configurations for all subsequent experiments** across different models, languages, attacks, and defenses. The Pareto curves themselves are also generated on this validation set to expose the behavior of each defense across different operating points. The test set is used **only once** to report the final evaluation results, and is never used for hyperparameter selection.

**Matched clean models.** We also clarify our evaluation protocol regarding matched clean models. Throughout the paper, **all defense comparisons are performed on matched clean models**. For every combination of model, language, and attack, all defenses are initialized from the **same poisoned checkpoint**, which is produced by poisoning the **same underlying clean ASR model**. Consequently, every defense is evaluated under identical starting conditions, ensuring that differences in the reported results arise solely from the defense method and its operating point, rather than differences in model initialization or training history.

**Corresponding updates in the revision: Section 4, Appendix K, and Figures 2 and 3.**